# Framework for the Machine Learning Based Wireless Sensing of the Electromagnetic Properties of Indoor Materials

Teodora Kocevska [1,2,†], Tomaž Javornik [1,2,*,†], Aleš Švigelj [1,2,†] and Andrej Hrovat [1,2,†]

1   Jožef Stefan Institute, Jamova Cesta 39, 1000 Ljubljana, Slovenia; teodora.kocevska@ijs.si (T.K.); ales.svigelj@ijs.si (A.Š.); andrej.hrovat@ijs.si (A.H.)
2   Jožef Stefan International Postgraduate School (IPS), Jamova 39, 1000 Ljubljana, Slovenia
*   Correspondence: tomaz.javornik@ijs.si
†   These authors contributed equally to this work.

**Abstract:** Available digital maps of indoor environments are limited to a description of the geometrical environment, despite there being an urgent need for more accurate information, particularly data about the electromagnetic (EM) properties of the materials used for walls. Such data would enable new possibilities in the design and optimization of wireless networks and the development of new radio services. In this paper, we introduce, formalize, and evaluate a framework for machine learning (ML) based wireless sensing of indoor surface materials in the form of EM properties. We apply the radio-environment (RE) signatures of the wireless link, which inherently contains environmental information due to the interaction of the radio waves with the environment. We specify the content of the RE signature suitable for surface-material classification as a set of multipath components given by the received power, delay, phase shift, and angle of arrival. The proposed framework applies an ML approach to construct a classification model using RE signatures labeled with the environmental information. The ML method exploits the data obtained from measurements or simulations. The performance of the framework in different scenarios is evaluated based on standard ML performance metrics, such as classification accuracy and F-score. The results of the elementary case prove that the proposed approach can be applied for the classification of the surface material for a plain environment, and can be further extended for the classification of wall materials in more complex indoor environments.

**Keywords:** indoor propagation; channel state information; ray tracing; decision tree; electromagnetic properties; radio environment signature; random forest; environment characterization; permittivity

## 1. Introduction

Environmental awareness enables many cutting-edge applications in a range of different domains, such as engineering, architecture, and construction. The potential uses of outdoor and indoor digital maps are in many different applications that will facilitate the human activities heading towards smart living in smart cities [1], such as spatial understanding (space planning, navigation, emergency response), automation (smart buildings, ambient intelligence applications, decision support, facility monitoring, elderly care), mediated reality (virtual reality, visualization, gaming), and wireless-system optimization.

The elevation maps of the Earth's surface, clutter maps, and building shapes, which are acquired using techniques, such as photogrammetry, light detection and ranging (LiDAR), interferometric synthetic aperture radar, land surveying, or geodetic building databases, provide sufficient information for a digital representation of the outdoor environment. However, the accurate description of the indoor spaces is still challenging due to the complex layout of the indoor structures including different architectural components and objects, such as furniture, equipment, etc. [2].

Indoor spaces are usually described with building plans that are often inaccurate, outdated, and difficult to obtain, or with reconstructed models of the environment from

point clouds. In both cases, these representations of the indoor environment include only the geometry of the building, without any semantic attributes of the indoor components. Some of the possible attributes that may be added to the accurate geometrical description are electromagnetic (EM) properties of the building materials and texture of the surfaces [3].

Such semantically enriched indoor maps are essential for many different applications. Since building structures and materials used indoors strongly affect the indoor radio propagation [4–6], the accurate indoor maps with material's EM properties can benefit to the design and optimization of the emerging wireless networks. Furthermore, radio-environment (RE) awareness is essential for development sixth generation (6G) wireless communication systems [7–9], which are looking at joint optimization of the RE and wireless networks by applying re-configurable intelligent surfaces [10]. Information about the RE can also decrease the extent of the training sequence and, thus, increase the link goodput. An awareness of the RE will result in improved wireless services and reduced environmental pollution with EM waves. The need for an accurate description of indoor spaces will be highlighted with near-future trends, such as increasing amounts of time spent indoors, mega-sized indoor structures in future cities, seamless and ubiquitous outdoor–indoor navigation, etc.

In this paper, we exploit the multipath propagation channel in indoor environments in order to sense the EM properties of the building materials by analyzing the RE impact on the received signal. In the literature, the approach is known as wireless or radio sensing. The radio signal propagating from the transmitter to the receiver is not directed only to the receiver location. Due to the nature of electromagnetic waves and antenna properties it can be broadcast in several directions. Misdirected radio signal may hit the object in the neighborhood. A part of the signal is reflected from the object, while the rest of it travels through the object, where it is additionally attenuated and delayed. The energy of the reflected signal depends on the electromagnetic properties of the material, i.e., the material conductivity, permittivity and permeability, and the roughness of the object surface. For example, highly conductive materials, such as metals and water, reflect the radio signals. On the other hand, building materials, such as brick, wood, and concrete, only partly reflect radio signal while part of it travels through them. Furthermore, when the object size is comparable or smaller to the signal wavelength, the object is invisible to the radio signal and does not obstruct the radio signal. The reflected radio signals may also hit the receiver antenna. Their path is longer compared to the path of the direct signal, which is expressed as a time delay and additional attenuation of the reflected signals [11]. Thus, the set of reflected signals with their delays and amplitudes is characteristic for the particular environment and is known as the radio-frequency or RE signature [12]. The RE signature can be presented as the channel transform function (CTF) [13], but in order to exploit the full potential of multipath propagation, we apply a set of the strongest multipath components described by received power, delay, phase shift, and angle of arrival (AoA) as the RE signature. Due to the problem complexity, we apply a data-driven instead of a model-driven approach to tackle the problem.

We focus on machine learning (ML)-based wall-material wireless sensing, which is modeled as a multi-class classification problem and solved by ML techniques. In particular, we proposed the complete framework applicable for different indoor environments and arbitrary transmitter and receiver layouts composed of several steps. First, a dataset of RE signatures for different wireless links in various propagation environments can be generated by applying a computer simulations or radio channel measurements [14].

The availability of suitable datasets is very limited, and the creation of such a dataset is very time consuming. Therefore, for the initial ML analyses, datasets obtained by computer simulations are used. Since ray-tracing algorithms are able to identify a set of the main most significant radio signal paths with their delays and attenuations between the transmitter and receiver, we apply them to generate RE signatures in a room where the material of the walls, the size and shape of the room, and the position of the transmitter and receiver vary. The generated dataset assumes a perfect transmitter and receiver, i.e., infinite

bandwidth, no additive noise, and linear and non-linear distortions in the transmit-receive chain. However, to investigate the impact of a certain non-ideality on the ML classification, the distortions in the transmit–receive chain, the noise and the constraints caused by the system bandwidth should be added later in a controlled manner. Next, the ML analysis on the dataset is performed in two steps. First, the learner is trained on labeled training data, i.e., RE signatures of radio links labeled with a target attribute representing the type of propagation environment. Second, ML models are applied to measured or simulated RE signatures and the target value, in our case the wall material, is predicted.

Based on the domain knowledge for radio-wave propagation we expect: (*i*) based on the electrical properties of materials some of them have a greater effect on the propagation than others, thus the environment information loaded in the RE signature is richer and consequently some materials can be more accurately classified than others; (*ii*) wireless links that correspond to some transmitter/receiver positions convey richer environmental information compared to others, and thus materials can be more accurately classified with some links than with others.

The main contributions of this study are as follows:

- We specified the RE signature for a rich multipath indoor environment, which is suitable for the classification of wall materials;
- We defined an ML-based framework for the estimation of a wall material which takes advantages of three engineering domains, i.e., measurements, modeling and simulations (environment and communication system modeling and ray-tracing simulations), artificial intelligence (ML techniques) and electromagnetics (EM properties of matter and radio-propagation mechanisms);
- With ML we confirmed the domain-knowledge-based assumptions, stating that the EM properties of the materials and the transmitter/receiver position affect the wall-material classification performance;
- We analyzed and compared four ML techniques for the wall-material classification applied to the same dataset.

The remainder of the paper is organized as follows. In the next section, the related work and the scientific background are given. The concept of the RE signature in the multipath channel is presented in Section 3. The description of the framework for ML-based indoor wireless sensing is described in Section 4. In Section 5, an example of the framework use case for indoor propagation scene is illustrated describing the complete framework usage procedure starting with definition of the propagation scenario, dataset building and analyzing the results. The paper is wrapped up with summary and directions for further work in the last section.

## 2. Background and Related Work

The methods for estimation of the EM properties of the indoor built-in materials based on ML techniques applied on RE signatures are not widely studied. However, in the literature there are several related topics, such as indoor mapping, wireless sensing, relative permittivity estimation, and indoor environment classification. The representative studies of the related topics are selected and summarized in the subsequent subsections in order to provide better understanding of the research work in different related domains and fine tune the direction of our study.

### 2.1. Indoor Mapping

The available maps of the indoor environments are: building plans in the form of two-dimensional (2D) paper drawings, three-dimensional (3D) digital models of the indoor space geometry, maps obtained by manual sites surveying or maps reconstructed from point clouds resulting from environment scanning with different technologies [15,16].

The manual surveying of indoor spaces is very time and man-power consuming, expensive, and of limited accuracy. The well-known examples of the approach are OpenStreetMap [17] where indoor spatial entities in public buildings, such as shopping malls,

office buildings, and airports have been tagged by volunteers, and MazeMap where public buildings, such as hospitals, universities, research campuses, etc., are mapped [18].

Another approach is 3D reconstruction of indoor environments by environment scanning. In [19], the publication statistics for the topic of 3D reconstruction of indoor environments was summarized and the state-of-the-art studies were reviewed and categorized according to their inherent principles. Environment scanning is two-step method where in the first step the distance to neighboring obstacles is measured in a set of predefined directions while in the second one the environment scans are converted to standard 3D objects and stored in standard computer formats [20]. In the first step light, sound or radio waves can be used. Most of the literature on environment scanning was devoted to the light-based approaches, LiDAR in [21,22], red, green, and blue (RGB) video combined with depth information [23], or infrared scanners [24,25]. The ultrasound-based approaches were not widely studied due to multiple reflections that occur from surrounding objects while the interest in still immature radio-based approaches is increasing [26–33]. Radio-based approaches are the most promising and of particular interest for the next-generation wireless communications. In fact, due to employing higher frequencies and consecutive reduced wavelength integration of large number of antennas and narrow electronically steering beams which greatly increase the mapping accuracy. In addition, the radio-based solutions can operate in non line-of-sight and poor visibility conditions.

### 2.2. Relative Permittivity Estimation

The relative permittivity estimation can be generally classified in relative permittivity estimation of material samples and relative permittivity estimation of materials built in varying indoor structures. Although the procedures for the individual materials were widely studied and reported in the literature, only few studies were dealing with the defining of the permittivity of heterogeneous indoor structures.

There exist several classes of methods for relative permittivity and permeability estimation of a material sample: free space measurement, transmission/reflection, and resonant. In the free space measurement methods, the relative permeability is estimated by observing the distortion introduced by a piece of material placed between transmitter and receiver [34]. In the transmission reflection methods, the relative permittivity is obtained by observing the reflection coefficient [35–39]. The resonant methods explore the resonant properties of the material [40] or the perturbation of a piece of material placed in the empty resonator. The latter approach is called also the resonant perturbation method. Recently, the cavity perturbation methods are particularly interested, where the original empty cavity is perturbed by the introduction of a small piece of dielectric [41].

Available approaches for relative permittivity estimation of material samples are not suitable for relative permittivity estimation of materials built in varying indoor structures. Thus, several in-situ measurements of the reflection coefficient within the environment of interest were conducted [42–46] where the reflection loss was used for the relative permittivity estimation [42,45,47–49]. This procedure results in accurate relative permittivity estimation of built-in materials which can be a tedious task when several surfaces materials in an indoor structure have to be covered. Therefore, in [3], the authors overcame the need for separate measurements of each material present in the environment by proposing the method which utilizes the multipath components from limited channel measurements to build a 3D permittivity map of the environment. To assure accurate permittivity estimates the method assumes that abundance of single-bounce specular reflections exist in the environment. The inputs of the method are accurate geometrical description of the environment in form of point cloud obtained with laser scanning, carrier frequency, locations of the transmitter/receiver antennas and set of multipath components for each transmitter–receiver radio link obtained with channel sounding. The reflecting surfaces were identified in the point cloud by matching the specular multipath components from measurements and ray tracing, while the relative permittivity for the points on the reflecting surface was obtained by solving inverse reflection problem.

## 2.3. Wireless Sensing

Wireless sensing is an emerging technique for acquiring information about a remote object and its characteristics without having any physical contact with it. It analyzes the impact of the remote object on the received acoustic, optical or radio signal. Other terms with similar meaning in the literature are device-free radio sensing, sensor-less sensing, radio imaging, non-invasive sensing, and zero-effort sensing. Early wireless-sensing methods estimated the signal strength on many different paths through the medium as radio tomographic imaging (RTI) [50], the round-trip time of the reflected signal (pulse radar) or phase (frequency-modulated continuous-wave radar), while the emerging approaches look at analyzing the channel state information (CSI) obtained from various wireless-communication systems. Analyzing the CSI in wireless sensing is referred to as radio analytics [51]. Recently, a huge effort has been put into taking advantage of wireless-communication technology, Wi-Fi, radio-frequency identification (RFID), mobile-communication systems, and wireless-sensor networks. The pros and cons of the different wireless technologies for sensing are analyzed in [52].

The wide deployment of Wi-Fi networks initiated research into Wi-Fi wireless sensing [53]. Ma et al. gave a comprehensive survey of Wi-Fi sensing, including signal processing techniques, algorithms, applications, challenges, and future trends [54]. The main approaches in Wi-Fi sensing, its limitations, application gaps in knowledge, and future directions were summarized in [55]. The studies looked at specific applications, such as intrusion detection, room-occupancy monitoring, daily-activity recognition, gesture recognition, vital-signs monitoring, user identification, and indoor localization and tracking [56,57].

The Internet of Things (IoT), based on wireless sensor networks (WSNs) [58], is the second approach to obtain information about the environment using wireless sensing. The large number of radio devices spread in an area of interest will communicate with each other, creating a huge mesh network. The obstacles between nodes additionally attenuate the radio signal and thus the environmental image can be estimated [50].

However, the application of WSNs to wireless sensing has several drawbacks, including costly deployment, device-location ambiguity, the power limitation of wireless nodes and their non-optimal location. In this respect, the robots equipped with odometers and gyroscopes were applied to environment mapping and localization. The approach was referred to as simultaneous localization and mapping (SLAM) [59,60].

In most applications the wireless sensing can be modeled as a pattern-recognition problem. Many traditional ML methods, such as decision tree (DT), random forest (RF), support vector machine (SVM), k-nearest neighbor (KNN) and deep learning, have been utilized to solve wireless-sensing problems [61]. Several studies have appeared recently investigating the possibilities of reducing the training and retraining, being the most labor-intensive tasks in ML-based wireless sensing. Different training options in a device-free radio frequency sensing system were discussed in [62]. The authors showed that reduced training might not necessarily kill the good performance, but some trade-offs will emerge. The feasibility of utilizing deep-similarity evaluation networks and collecting samples with deep generative adversarial networks for reducing the training efforts were studied in [61].

## 2.4. Indoor Environment Classification

A methodology for the ML classification of indoor environments based on CTF and the frequency coherence function (FCF) was proposed by AlHajri et al. [63]. They investigated how CTF and FCF vary within the room and they proved that FCF and CTF can be considered as a unique fingerprint of the environment. In their later work, they explored several ML classification methods using different combinations of metrics as features [12,13]. Their findings are limited to the classification of four different environments with different levels of clutter: no clutter, low clutter, medium clutter, and high clutter.

Compared to the methods presented by AlHajri et al., in our work we apply a more complex RE signature using the channel impulse response (CIR) enreached by AoA of each signal component, while they apply only CTF and its auto-correlation. Furthermore, we look at predicting the EM properties of a building material used for a particular wall, while AlHajri et al. were looking for how cluttered the environment is in a room, considering the environment as a whole and not paying attention on the size of the room and the type of the materials forming the walls of the room.

## 3. Radio-Environment Signature

In a rich, multipath, indoor environment, the transmitted signal interacts with the surfaces [64]. When the signal collides with a surface, a part of the signal is reflected and several copies of the transmitted signal reach the receiver via different paths. The wireless multipath channel is formed of signal paths differing in terms of signal amplitude, delay, phase shift, and AoA. A set of signal paths compose the CSI. A wireless link between the transmitter–receiver (Tx-Rx) with omni-directional antennas in an indoor environment is depicted in Figure 1. The EM waves are represented as radio rays. The yellow stars show the rays' interactions with the surfaces. Surface roughness influence the ratio between the specular reflection and scattering components. The incidence angle impacts the degree of scattering and the roughness of the surface seen by radio waves is reduced with closing incidence angle to grazing incidence. The line of sight (LoS) path is denoted by a red line, single bounce reflections are illustrated by green lines and the scattering is represented with blue lines.

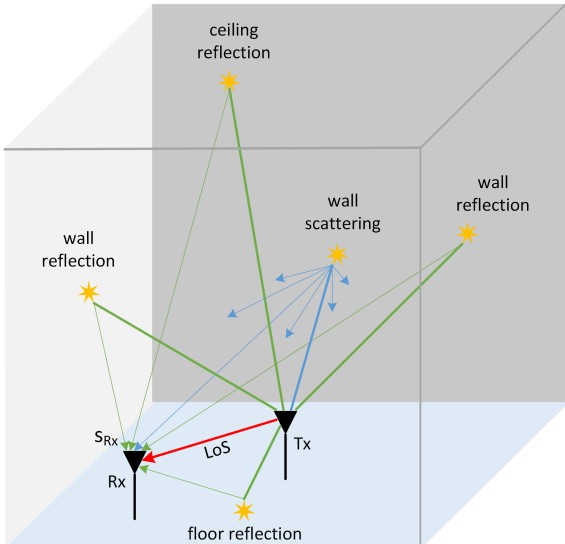

**Figure 1.** Multipath component propagation channels between Tx-Rx in an indoor environment.

As a result of the multipath propagation, the received signal $s_{Rx}$ is a superposition of multiple replicas of the transmitted signal described by

$$s_{R_x} = \sum_{i=0}^{\infty} s(P_i, \tau_i, \phi_i, \theta_i, \varphi_i),\tag{1}$$

where $P_i$, $\tau_i$, $\phi_i$, $\theta_i$ and $\varphi_i$ are the power, delay, phase shift, and the AoA azimuth and AoA elevation of the signal multipath component $i$, respectively.

Since the propagation environment affects the signal propagation, the information regarding the environment is included in the radio signals. Thus, each particular environment setting results in a corresponding propagation-distortion pattern. Even small changes to the environment affect the signal propagation, which enables an estimation of the environment's characteristics, i.e., geometry, EM properties of the surface materials, etc. Rich spatial information is contained in the signal with a large number of multipath components. The number of multipath components that can be distinguished depends on the signal bandwidth, i.e., a large bandwidth results in a good spatial resolution [65]. The set of $P_i$, $\tau_i$, $\phi_i$, $\theta_i$ and $\varphi_i$ for all the multipath components arriving at the Rx in position $B$ from the Tx in position $A$ is a partial RE signature $R_{AB}$ that corresponds to that particular Tx-Rx positioned in $A$ and $B$ and can be written as

$$R_{AB} = \{[P_i, \tau_i, \phi_i, \theta_i, \varphi_i]\}_{i=0\rightarrow\infty},\tag{2}$$

where the delay $\tau_i$ is a function of the propagation distance, i.e., $\tau_i = d_i/c_0$, $d_i$ is the run length of the $i$-th path and $c_0$ is the speed of light.

Different wireless links contain different amounts of environment information, depending on the interaction with the environment. However, the surrounding environment cannot be estimated from a single $R_{AB}$ that corresponds to a single wireless link. Hence, a set of partial RE signatures obtained from a set of wireless links, i.e., (Tx-Rx)s placed at different positions in the indoor environment, are needed. The set of partial RE signatures referred to as the RE signature is defined by

$$R = \{R_{AB}|A, B \in I\}, I = \{1, 2, \ldots, \infty\}.\tag{3}$$

The RE signature depends on the configuration of the wireless system and the surrounding environment. Since the configuration of the communication system is known, it can be assumed that the surrounding environment, together with the material properties, can be estimated from the RE signature by correlating the environment with the corresponding distortion pattern.

## 4. ML-Based Wireless-Sensing Framework

To streamline the conducted procedure, we developed a framework that enables the design and evaluation of the ML-based sensing of EM material properties in an indoor environment. The framework architecture is given in Figure 2.

The proposed framework is composed of the following modules:

- RE acquisition module: collects the propagation parameters of multiple wireless links in different indoor environments by measurements or computer simulations, pre-processes the results and transforms them into the form of partial RE signature;
- Propagation characteristics storage module: stores the partial RE signature, environment and radio system description in a database for the purposes of building a large dataset for the ML task and as open-access data for other RE-related studies.
- ML-based radio-analysis module: builds training and testing datasets from the main dataset according to a predefined scenario, extracts knowledge from the training set, builds a model and applies the model to classify the RE from the input RE signature.
- ML performance-evaluation module: calculates ML performance metrics and stores the statistics.
- Domain-knowledge-based interpretation module: Evaluates the classification results by taking into account the domain knowledge in the field of EM propagation.

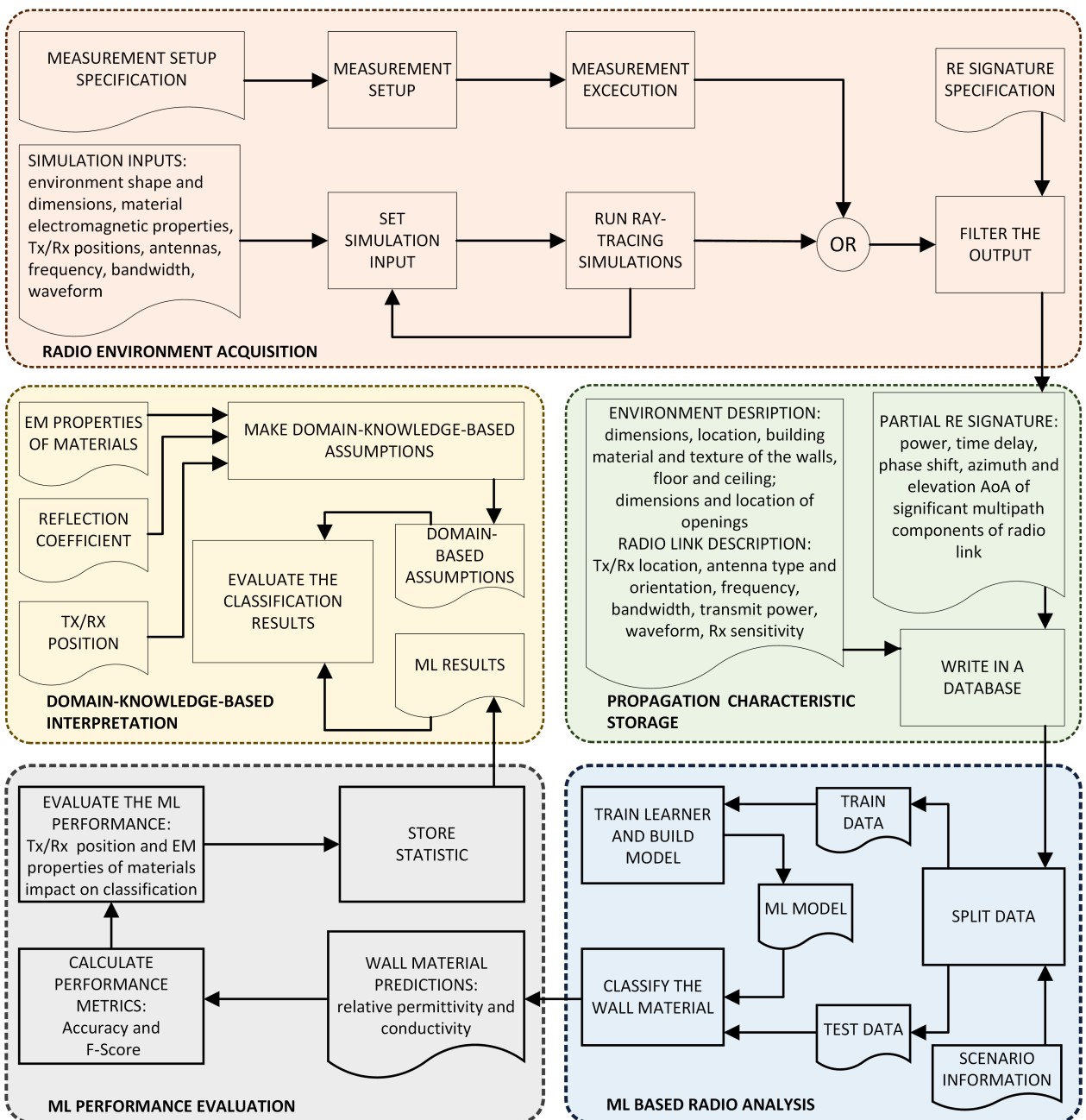

**Figure 2.** Wireless sensing framework architecture.

## 4.1. Radio-Environment Acquisition Module

The RE acquisition module acquires the partial RE signatures of multiple wireless links that correspond to predefined transmitter and receiver positions in various indoor environments specified by the given shape, dimensions, and EM properties of surface materials. The partial RE signature can be obtained experimentally by radio-channel measurements or by computer simulations. In either cases the system setup must be specified (used technology, environment, Tx-Rx positions, frequency, bandwidth, antennas, transmit power, output parameters, and measurement procedure). When performing the measurements, the equipment must be setup at the chosen location, measurements must be executed according to the predefined scenario and the result stored, filtered and forwarded to the database. Computer simulations require to set predefined simulations inputs, run ray-tracing engine, adapt the ray-tracing simulation parameters if needed and, similar to measurements, collect, filter, and forward the output to the database.

The channel sounding is widely used for obtaining CIR, CSI, and RE signature. The transmitter transmits a predefined signal, which is detected by the receiver. CIRs are calculated using the knowledge of the transmitted and received signals. The received signal contains contributions from other sources that transmit at the same frequency band and the additive Gaussian noise. A noisy CIR can have a huge impact on the performance of ML algorithms. The sounding signal has to fulfill several requirements, such as a large bandwidth, a uniform spectral density across the bandwidth, etc., to achieve sufficient accuracy of the CIR. Since the channel sounding requires professional, high-cost radio equipment, it will rarely be applied for a massive RE signature estimation. An alternative approach for an experimental estimation of CIR is using off-the-shelf radio equipment, which supports the estimation and recording CIR. For example, the radio equipment based on the IEEE Standard for Local and Metropolitan Area Networks [66], using ultra-wideband (UWB) communication technology supports mechanisms to estimate the CIR from a transmitted training sequence [67–69].

Computer simulations offer a controllable environment for obtaining the RE signature in various indoor and outdoor environments. Ray-tracing is usually considered as an efficient way to obtaining the RE signature [14,70]. It is a tractable, brute-force method for calculating the progress of wave fronts [71]. It is a powerful approximation approach that is less computationally demanding compared to methods based on Maxwell's equations. Its accuracy depends mainly on the ratio of the wavelength to the dimensions of the interfering objects. The most accurate estimations are achieved when the interfering objects are large compared to the wavelength. Any ray-tracing software regarding the used principle for tracing the rays, i.e., ray shooting and bouncing (SBR) or method of images, can be used as long as it provides adequate outputs needed for further processing [72]. Computer simulations are particularly convenient when all the aspects that affect the propagation have to be controlled.

### 4.2. Propagation Characteristic Storage Module

In order to support ML-based radio analysis and reuse of the measured or simulated CSI, the following information is stored in a database:

- Description of the environment;
- Description of the radio system, (i.e., transmitters and receivers);
- RE signatures.

In general, any publicly available file format can be applied to store information about the indoor environment. However, to make the indoor description suitable as inputs to different indoor radio-propagation software and 3D representation, we present an indoor environment as a set of cuboids (walls), which have one or more openings (holes). The information about the wall includes its dimensions (length, width, height), its location (translation and rotation), the material it is made of and information about the texture that might have an impact on scattering of the radio signal. The information about the opening consists of the opening's size and its position relative to the wall's origin [73].

The description of the radio system includes the position of transmitter and receiver, as well as the type and orientation of the antenna. The transmit power, antenna type, carrier frequency, signal bandwidth, and transmitted-signal waveform are added for the transmitter. In order not to look for a very weak signal, the receiver sensitivity is added for the receiver.

The proposed framework presents a general solution and, as such, is able to consider wide variety of parameters as RE signature since it is designed not to be limited to specific use cases, measurement equipment capabilities neither to radio propagation simulation tool. Thus, depending on the capabilities and requirements the suitable subset can be defined. The radio channel description, the RE signature, can contain a set of received radio-signal components (rays) with the following information: $P_i$ received power of the received signal component, $\tau_i$ delay of the signal component, $\phi_i$ phase shift of signal component, $\varphi_i$ AoA elevation and $\theta_i$ AoA azimuth. The number of signal components

is not hard limited and depends on the energy of the components. The contribution of the components with the higher number of reflections is, due to very low energy, usually negligible, and thus is not stored in the dataset. The database can be implemented as a separate entity, as an open-access database, or it can be a part of the framework.

### 4.3. Machine Learning Based Radio-Analysis Module

In the ML-based radio-analysis module, first, the main dataset is built which is then split on train and test dataset. Next, a machine learning algorithm is run on the training data, a model is built and saved for the later use. Finally, the model is used for making predictions on unseen data samples from the test dataset.

The main dataset is built as a selection of propagation data of radio links stored in the database. The selection is made based on selection of indoor spaces, radio links, and RE signature structure. The dataset is a collection of samples, where each sample contains propagation data that corresponds to single radio link defined with the positions of the Tx and Rx. The number of input attributes depends on the number of propagation parameters in the RE signature structure and the number of considered rays. The dataset is labeled with a single multi-class target attribute that represent the surface material. Meaning that each sample can be assigned to only one class of finite set of material categories. The input attributes are numeric, while the output attribute is nominal categorical. The ML task belongs to supervised learning and can be best framed as multi-class single-target classification. It predicts the surface material class as target based on the link RE signature as input.

Since the amount of available data are limited, the main dataset is split to non-overlapping train and test subsets according to predefined scenario. The defined splitting scenario has to reflect the real world situation under investigation and training, additionally the testing dataset must contain representative data of the underlying problem. The generality of the proposed framework enables the consideration of a variety of splitting scenarios.

The task of multi-class classification is widely studied and many ML algorithms are found to be suitable for solving it [74,75]. Considering the underlying problem we selected a shortlist of machine learning algorithms that can be used for surface material prediction. When creating the shortlist of algorithms the nature of the data, interpretability, and the predicting performance were taken into account. The simpler algorithms are expected to provide understandable models allowing us to discover the most important correlation between the input and output attributes. On the other hand, more complex algorithms are expected to provide high predicting performance with less interpretable models.

The selected shortlist includes simple algorithms, such as DT and Naive Bayes (NB), as well as more complex algorithms, such as RF and Multilayer Perceptron (MP) [76–79]. NB is generative probability technique used for classification problems which is based on Bayes Theorem. The major assumption of NB is that all features are mutually independent. It can capture patterns in the data and provide efficient models with short training time. Furthermore, it does not require large training data, thus it can be considered as a starting point for analysing practical datasets and obtaining the base accuracy of the dataset. DT models are obtained without large computational expenses even on large datasets. The models are interpretable and easy to understand. This algorithm is suitable for inferring the correlations between input and target variables, visualizing how the learner extracts knowledge from the data and for problems that require a short run-time. The predictive performance of DTs can be further improved if they are combined into an ensemble. An ensemble combines the predictions of a set of base models to obtain an overall prediction. The performance of the ensemble will be higher if the base models have high performance and are uncorrelated. RF is one of the most famous ensemble methods for tree based predictive models that can improve the performance of DT and prevent from overfitting. MPs provide models through highly interconnected perceptrons that accept inputs, apply weighting coefficients and feed their output to other perceptrons which continue the

process through the network to the output. MPs involve constant back and forth until the error is minimized, state known as convergence. MPs can learn from extremely large datasets and provide models with high predictive performance. However, these models are hard to understand. The parameters of DT, RF and MP are summarized in Table 1.

**Table 1.** Summary of the parameters used for DT, RF, and MP [80].

| ML Algorithm | Parameter | Description | Value |
|---|---|---|---|
| DT | Confidence factor | Confidence factor used for pruning | 0.25 |
| | Number of objects | Minimum number of instances per leaf | 2 |
| | Number of folds | Size of the pruning set | 3 |
| RF | Number of trees | Number of trees in the forest | 25 |
| | Number of instances | Minimum number of instances per leaf | 1 |
| | Number of levels | Maximum number of levels in each decision tree | unlimited |
| MP | Learning rate | Learning rate for the backpropagation algorithm (value between 0 and 1) | 0.3 |
| | Momentum | Momentum rate for the backpropagating algorithm (value between 0 and 1) | 0.2 |
| | Traininng time | Number of epochs to train trough | 500 |

*4.4. Machine Learning Performance-Evaluation Module*

In the ML performance-evaluation module, the performance of the classifier is evaluated with ML performance metrics. The results are first stored in a database and next used in the domain-knowledge-based interpretation module.

An evaluation of the ML algorithm is an essential part in the ML pipeline. The standard performance metrics calculated from the confusion matrix (CM) are applied for the evaluation [81]. The CM describes the complete performance of the model. The rows represent the true labels of the instances and the columns represent the predicted labels in the CM. In the multi-class classification problem, the element $E_{ij}$ of the CM represents the number of instances that belong to class $C_i$ and are classified as class $C_j$, $i, j = 1, 2, \ldots, N_c$, where $N_c$ is the number of classes. Each prediction falls into four categories denoted as true positives ($TP$), true negatives ($TN$), false positives ($FP$), and false negatives ($FN$). The performance metrics suitable for an environment-classification task are:

- Classification accuracy is the ratio between the number of correct predictions and the total number of input instances. It is a relevant metric in our classification problem since the number of instances belonging to each class is the same. The classification accuracy for the *i*th class is calculated with (4):

$$Accuracy_i = \frac{TP_i + TN_i}{TP_i + TN_i + FP_i + FN_i}. \tag{4}$$

- F-score (*F*) is the harmonic mean of precision and recall. It is the measure of classifier precision and robustness. It tells us how many instances are classified correctly and if it does not miss a significant number of instances. The large *F* values represent good performance of the model. *F* calculated for the *i*th class is obtained by using (5)–(7):

$$Precision_i = \frac{TP_i}{TP_i + FP_i}, \tag{5}$$

$$Recall_i = \frac{TP_i}{TP_i + FN_i}, \tag{6}$$

$$F_i = 2 \times \frac{Precision_i \times Recall_i}{Precision_i + Recall_i}. \tag{7}$$

### 4.5. Domain-Knowledge-Based Interpretation Module

Since ML techniques are intended for general purpose, the use of domain knowledge is valuable for benchmarking the ML results and validating that the conclusions are in line with the assumptions based on existing physical models. In our study, the classification results can be interpreted with those properties of the RE that affect the signal propagation. Assuming the same geometry of the RE, the important role is played by the EM properties of the wall materials. The fundamental EM property of the material is the relative permittivity $\epsilon_r$. Different related quantities, such as refractive index, loss tangent, etc., depend on it, assuming the signal carrier frequency is known [4]. Permittivity is a dimensionless, complex-valued quantity (8). Its real and imaginary parts are denoted as $\epsilon_r'$ and $\epsilon_r''$, respectively:

$$\epsilon_r = \epsilon_r' + j\epsilon_r''. \tag{8}$$

Conductivity $\sigma$ is related to the imaginary part of $\epsilon_r$ as given in (9) where $\epsilon_0$ is the permittivity of free space ($8.854 \times 10^{-12}$ F/m) and $\omega_c = 2\pi f_c$ is the angular frequency in rad/s.

$$\epsilon_r'' = \frac{\sigma}{\epsilon_0 \omega_c} \tag{9}$$

The reflection coefficient of a material is a function of the relative permittivity of the material $\epsilon_r$ and the angle of incidence $\alpha$. For vertical ($VP$) and horizontal polarization ($HP$) it can be calculated according to (10) and (11), respectively, where the angle of incidence is equal to the angle of reflection [82]:

$$\Gamma_{VP} = \frac{\epsilon_r \cos\alpha - \sqrt{\epsilon_r - \sin^2\alpha}}{\epsilon_r \cos\alpha + \sqrt{\epsilon_r - \sin^2\alpha}} \tag{10}$$

and

$$\Gamma_{HP} = \frac{\cos\alpha - \sqrt{\epsilon_r - \sin^2\alpha}}{\cos\alpha + \sqrt{\epsilon_r - \sin^2\alpha}}. \tag{11}$$

The reflection coefficients as functions of the incidence angle for brick, concrete, glass, and wood are plotted in Figure 3. It shows that the reflection coefficient of the material depends on its relative permittivity and the angle of incidence. The materials with similar permittivities, for example, concrete and glass, have similar reflection coefficients and, thus, they have similar impacts on the RE signature, as seen from Table 2 and Figure 3. Materials with a large difference in the relative permittivity, for example, wood and glass, perform completely differently. However, when the angle of incidence is larger than 80 degrees, i.e., the radio ray become nearly parallel with the facet of obstacle, the reflection coefficient approaches the same value, i.e., 0 dB, and all materials perform similarly, even they have a large difference in relative permeability. However, in an indoor environment, the incidence angle is rarely above 80 degrees, since the transmitter and receiver are usually not placed close to the walls.

Given the reflection coefficient of the materials, it is expected that the ML algorithms will distinguish all the classes. However, we expect that the separation between some classes is clearer than between the others. In particular, the separations between wood and glass, wood and concrete, and brick and glass should be very clear, while the algorithms could face a problem in distinguishing concrete from glass, brick from concrete, and wood from brick.

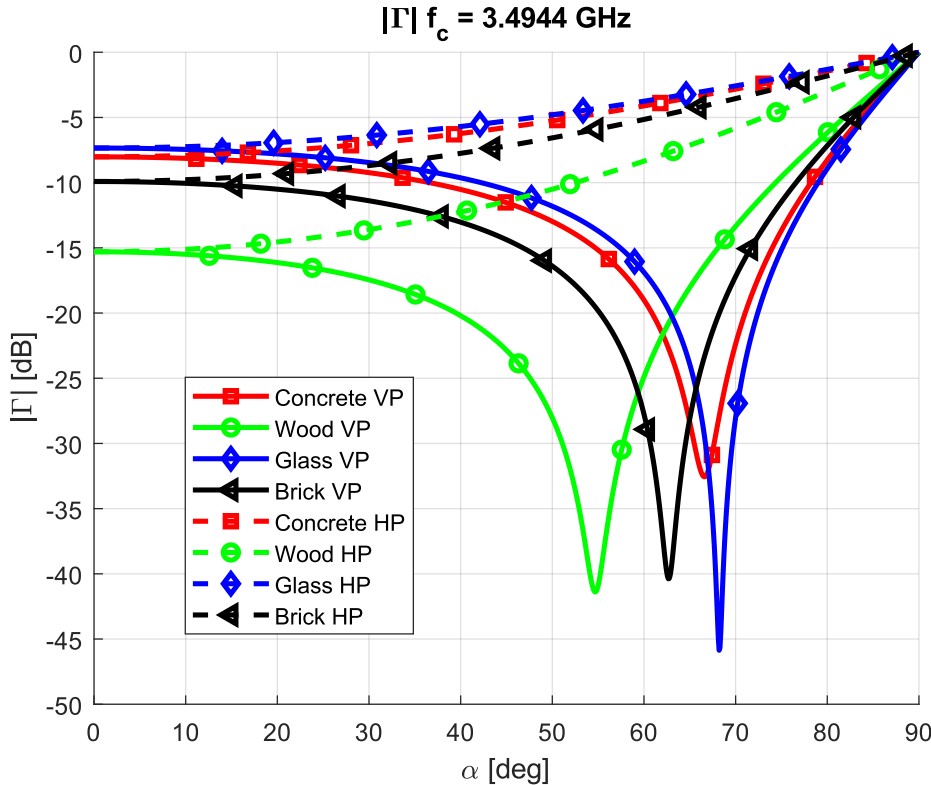

**Figure 3.** Comparison of the reflection coefficients of brick, concrete, glass, and wood.

**Table 2.** Electrical properties of wall materials [4].

| Material | $\epsilon_r'$ | $\sigma$ (S/m) |
|---|---|---|
| Brick | 3.75 | 0.038 |
| Concrete | 5.31 | 0.120 |
| Glass | 6.27 | 0.029 |
| Wood | 1.99 | 0.026 |

## 5. Framework Use Case Example

The proposed framework is evaluated by initial scenario where the EM material of a single wall in a room, without any doors, windows, and furniture is classified. We applied computer simulations (i.e., ray-tracing) to build the dataset, specified the train–test scenarios and discussed and evaluated the obtained results. The initial indoor scene setting and the computer simulations for obtaining the RE signatures were selected in order to test the applicability of the proposed concept and to analyze the impact of the EM properties of the materials and Tx-Rx position on the classification performance.

### 5.1. Dataset Building

Building the RE signature dataset consists of the following steps:
- Setting up an indoor scene, including the geometry and EM properties of the wall materials;
- Specification of the communication technology and its parameters and the location of the radio nodes;
- Specification or ray-tracing algorithm;
- Execution of simulations to obtain the dataset.

The indoor scene is an empty room with no doors and windows. The room has a square floor with size 3 m × 3 m. The ceiling is 2.5 m above the floor. The room is constrained with six surfaces: floor, ceiling, wall 1 (W1), wall 2 (W2), wall 3 (W3), and wall 4 (W4). The floor,



ceiling, walls W1, W2, and W3 are made of concrete. The material of wall W4 should be classified, and thus it varies for each simulation run. The electrical properties of the considered wall materials are summarized in Table 2 in terms of $\epsilon'_r$ and $\sigma$ [4]. The materials are considered to be non-ionized and non-magnetic; therefore, the free-charge density $\rho$ is set to zero and the permeability for all the materials $\mu$ is set to the permeability of free space ($\mu_0 = 4\pi \times 10^{-7}$ H/m).

The parameters of the communication technology were specified in such a way that the study case can be upgraded with the experimentally measured RE signatures in a real environment. In this respect, the central frequency $f_c$ is 3494.4 MHz and the bandwidth *BW* is 499.2 MHz, which corresponds to one of the channels supported by UWB radio communications according to the IEEE802.15.4-2011 UWB standard [66,69]. At the transmitter and receiver, omni-directional antennas are considered. Both the transmitter and receiver antennas are 2 m above the floor. The positions of the transmitter and receiver were determined to uniformly cover the floor of the room, no transmitter overlaps any receiver, different distances of the transmitters from the walls and to capture the most realistic, as well as corner cases. In this respect, we selected seven transmitter positions, one at the room's center, others almost equally distributed around the central transmitter. The coordinates of the transmitters are summarized in Table 3 and the transmitter position with appropriate labels, i.e., CC, DC, DL, DR, UC, UL, and UR, corresponding to the center, down-center, down-left, down-right, upper-center, upper-left, and upper-right, are illustrated in Figure 4 with blue stars. The 676 receivers were laid out on a square 2-D grid with 26 horizontal and 26 vertical grid lines, forming squares with a size of 0.1 m × 0.1 m. Receivers were positioned at the grid lines' intersection lines and are depicted in Figure 4 with orange dots. The corners of the receiver grid were placed 0.25 m from the walls.

**Table 3.** The coordinates of transmitters.

| Transmitter Position | Position Coordinates (x,y) |
| --- | --- |
| CC | (1.5 m, 1.5 m) |
| DC | (1.5 m, 0.1 m) |
| DL | (0.5 m, 0.5 m) |
| DR | (2.5 m, 0.5 m) |
| UC | (1.5 m, 2.9 m) |
| UL | (0.5 m, 2.5 m) |
| UR | (2.5 m, 2.5 m) |

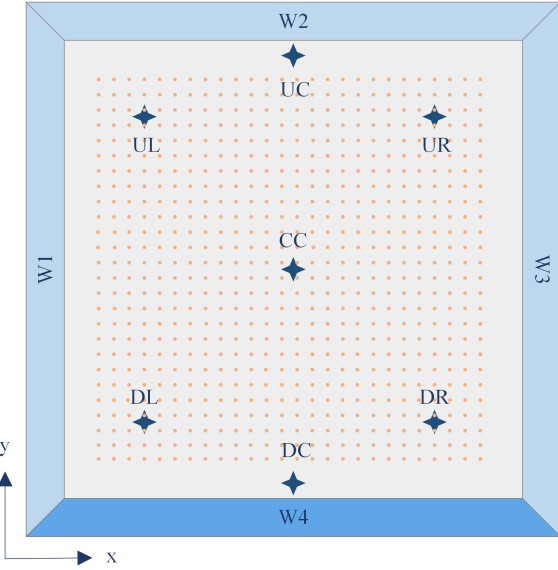

**Figure 4.** Transmitters (stars) and receivers (dots).

There are many software programs for radio-channel prediction that use ray tracing [73,83]. In our case, the ray-tracing simulations are conducted using the commercially available software Remcom Wireless InSite, which is capable of evaluating the site-specific propagation characteristics between the Tx-Rx and of generating accurate values for specific propagation parameters [83]. Wireless InSite is a suite of ray-tracing models and high-fidelity EM solvers. It has several functionalities that are particularly suitable for our study. First, its modeling and importing functionality allows the environment and communication system entities to be modeled within the software, as well as imported from available databases. Next, the command-line execution functionality allows the use of scripts for input parameter values' change and execution in the case of exhaustive simulations. Finally, all the output is saved in ASCII format files that can be post-processed externally.

The propagation was modeled with the implementation of a full 3-D model. The SBR method takes account of the 3-D surrounding geometry that was used [84]. Since the analyzed scene is an empty room with no edges where the diffraction could emerge and the receiver are placed within a room the diffraction and transmission effects are not present and thus have no need to be considered. For this reason, only reflections were considered when building the dataset. The closed room scene generates an infinite number of multipath components, but after several reflections the multipath component level reaches the receiver sensitivity.

Transmitters at seven positions together with receivers at 676 positions form 4732 wireless links, which, when combined with four different wall materials, result in 18,928 partial RE signatures. Since a large dataset of a RE signature has to be built using the ray-tracing tool, we developed a Python script for the automatic change of the simulation input parameters, including the environment shape, the environment dimensions, such as width, length, and height, the EM properties of the surface materials, the wireless system's configuration parameters, such as waveforms and antenna properties, the transmitter and receiver positions.

A study-case dataset is built from the database as a selection of observed environments, radio links, and a RE signature corresponding to the radio link. In the presented use case example the dataset conveys the propagation distortion patterns corresponding to the different materials for all the observed radio links.

### 5.2. Dataset Summary

The dataset is a matrix of instances versus attributes. It has 18,928 rows and 126 columns. The number of instances corresponds to 4732 radio links obtained in four rooms with different wall materials. Each instance represents a RE signature of a single wireless link in an analyzed environment. The dataset has 125 numerical input attributes representing the received power, delay, phase shift, AoA elevation, and AoA azimuth for the 25 strongest radio rays between pair of radio nodes ($P_1, \tau_1, \phi_1, \theta_1, \varphi_1, \ldots, P_{25}, \tau_{25}, \phi_{25}, \theta_{25}, \varphi_{25}$). The data are labeled with nominal target attributes, that describe the surface material and can catergorized as one of four categories (brick, concrete, glass, wood). An extract of the dataset is shown in Figure 5. The dataset is balanced, i.e., all class categories have the same number of observations and the number of instances per class is 4732.

| index | power$_1$ | delay$_1$ | shift$_1$ | azimuth$_1$ | elevation$_1$ | $\cdots$ | power$_{25}$ | delay$_{25}$ | shift$_{25}$ | azimuth$_{25}$ | elevation$_{25}$ | class |
|---|---|---|---|---|---|---|---|---|---|---|---|---|
| 0 | $-36.01$ | $1.89 \times 10^{-9}$ | 167.10 | 355.94 | 90.00 | $\cdots$ | $-74.52$ | $2.08 \times 10^{-8}$ | 131.77 | 284.21 | 90.00 | concrete |
| 1 | $-48.46$ | $7.91 \times 10^{-9}$ | 145.41 | 124.27 | 90.00 | $\cdots$ | $-75.33$ | $1.55 \times 10^{-8}$ | 15.05 | 124.27 | 149.33 | brick |
| 2 | $-48.10$ | $7.59 \times 10^{-9}$ | $-111.10$ | 83.33 | 90.00 | $\cdots$ | $-77.74$ | $1.40 \times 10^{-8}$ | 159.68 | 37.05 | 76.21 | wood |
| 3 | $-43.53$ | $4.49 \times 10^{-9}$ | 28.10 | 19.99 | 90.00 | $\cdots$ | $-81.01$ | $1.99 \times 10^{-8}$ | 26.42 | 230.55 | 80.44 | wood |
| 4 | $-43.10$ | $4.27 \times 10^{-9}$ | $-12.88$ | 318.94 | 90.00 | $\cdots$ | $-77.28$ | $2.71 \times 10^{-8}$ | 149.96 | 30.86 | 90.00 | wood |
| 5 | $-39.74$ | $2.90 \times 10^{-9}$ | 150.23 | 196.02 | 90.00 | $\cdots$ | $-73.01$ | $1.71 \times 10^{-8}$ | 98.47 | 220.23 | 78.77 | glass |
| $\vdots$ | $\vdots$ | $\vdots$ | $\vdots$ | $\vdots$ | $\vdots$ | $\cdots$ | $\vdots$ | $\vdots$ | $\vdots$ | $\vdots$ | $\vdots$ | $\vdots$ |

**Figure 5.** Excerpt from dataset.

### 5.3. Simulation Scenarios

In order to investigate how the transmitter position affects the wall-material classification, we specified three scenarios, which reflect the actual real world situations, as well as provide good inside of the results, as follows:

- $C_{train}C_{test}$: training and testing on the dataset that corresponds to the central transmitter position;
- $M_{train}M_{test}$: training and testing on the dataset that corresponds to multiple transmitter positions;
- $C_{train}M_{test}$: training on the dataset that corresponds to the central transmitter position and testing on the data that corresponds to the multiple non-central transmitter positions.

In the $C_{train}C_{test}$ scenario the training data corresponds to the CC transmitter position and the 96 random receiver positions, while the testing data corresponds to the same transmitter position used for the training and the remaining 580 receiver positions. In the $M_{train}M_{test}$ scenario, the training data corresponds to the CC, DC, DL, DR, UC, UL, and UR transmitter positions and the 96 random receiver positions, while the test dataset corresponds to the same transmitter positions used for the training and the remaining 580 receiver positions. In the $C_{train}M_{test}$ scenario, the training dataset corresponds to the CC transmitter position and all 676 receiver positions, while the test data correspond to multiple different transmitter positions, i.e., DC, DL, DR, UC, UL, and UR and all 676 receiver positions.

### 5.4. Results and Analysis

In order to evaluate the performance of the framework, we calculated the confusion matrices for all three scenarios. The results are presented in Figure 6. The darker-colored cells of the matrix represent higher percentage values. The dark-matrix diagonal elements and light off-diagonal elements represent good model performance. In this respect, the models based on the $C_{train}C_{test}$ and the $M_{train}M_{test}$ scenarios perform significantly better than the models based on the $C_{train}M_{test}$ scenario. The reason is that in the $C_{train}M_{test}$ scenario the learner was trained using only radio links with a single transmitter position, while it was tested on radio links, where the transmitter position is significantly different from the training links. Further, by observing the confusion matrices for different algorithms for scenario $C_{train}M_{test}$ it seems the RF shows better performance than the other tested ML algorithms.

Thus, in the next step we compared the prediction performance of selected ML algorithms when applied to the same data regarding the percentage of correctly classified instances. It is demonstrated in the Table 4, where the accuracies of the NB, DT, MP, and RF in different scenarios are given, that all algorithms have more than 68% correctly classified instances for each scenario. Additionally, it can be seen that NB has lowest and RF highest percentage of correctly classified instances in all scenarios, respectively. For scenario $C_{train}C_{test}$ all algorithms have more than 94% correctly classified instances and the difference in performance between algorithms is less than 5%. For scenario $M_{train}M_{test}$ DT and MP perform similar to RF while NB has around 13% less correctly classified instances. RF significantly outperforms all algorithms in scenario $C_{train}M_{test}$. Additionally, it should be noted that while the difference in correctly classified instances between RF and MP for scenarios $C_{train}C_{test}$ and $M_{train}M_{test}$ is very small, i.e., less than 1%, for scenario $C_{train}M_{test}$ the difference is significantly larger, i.e., around 14%. These results indicate that for scenario $C_{train}C_{test}$ all algorithms can be used, for scenario $M_{train}M_{test}$ DT, MP, and RF should be preferred over NB, and for scenario $C_{train}M_{test}$ RF should be preferred over other tested algorithms.

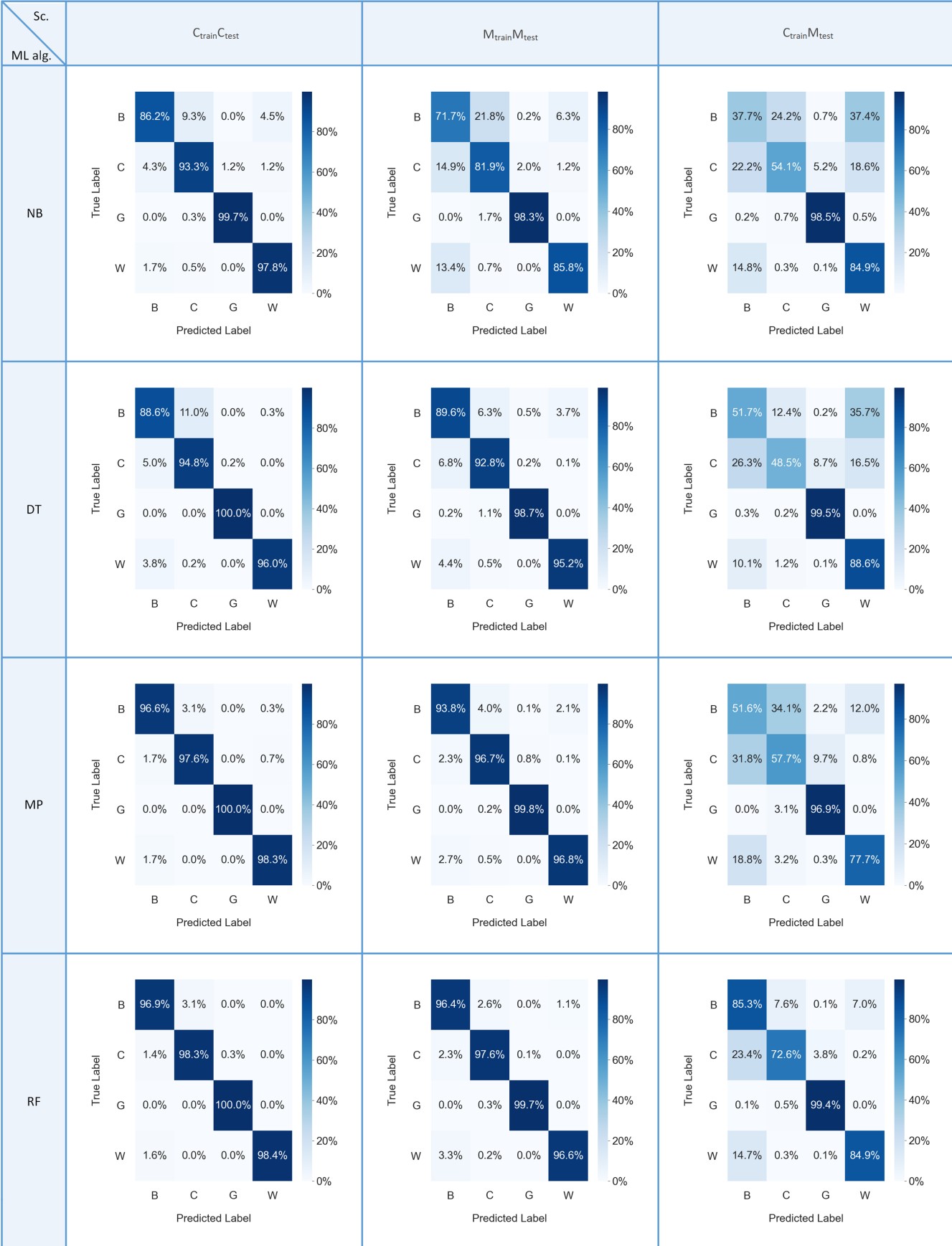

**Figure 6.** Confusion matrices for different ML algorithms and scenarios, (B—brick, C—concrete, G—glass, W—wood).

**Table 4.** Accuracy of DT and RF in different scenarios.

| Scenario | ML Algorithm | | | |
|---|---|---|---|---|
| | NB | DT | MP | RF |
| $C_{train}C_{test}$ | 94.2% | 94.9% | 98.1% | 98.4% |
| $M_{train}M_{test}$ | 84.4% | 94.1% | 96.8% | 97.6% |
| $C_{train}M_{test}$ | 68.8% | 72.1% | 71.0% | 85.5% |

We calculated F-score since for the material-classification task we would like to maximize the $TP$s, while the $FP$s and $FN$s are both costly. Table 5 summarizes the $F$ values per class for all three scenarios and for all ML algorithms, calculated on the basis of the confusion matrices given in Figure 6.

**Table 5.** F-score per material with different ML algorithms.

| Material | Scenario | | | | | | | | | | | |
|---|---|---|---|---|---|---|---|---|---|---|---|---|
| | $C_{train}C_{test}$ | | | | $M_{train}M_{test}$ | | | | $C_{train}M_{test}$ | | | |
| | ML Algorithm | | | | | | | | | | | |
| | NB | DT | MP | RF | NB | DT | MP | RF | NB | DT | MP | RF |
| Brick | 0.90 | 0.89 | 0.97 | 0.97 | 0.72 | 0.89 | 0.94 | 0.96 | 0.43 | 0.55 | 0.51 | 0.76 |
| Concrete | 0.92 | 0.92 | 0.97 | 0.98 | 0.80 | 0.93 | 0.96 | 0.97 | 0.60 | 0.60 | 0.58 | 0.80 |
| Glass | 0.99 | 0.99 | 1.00 | 0.99 | 0.98 | 0.99 | 0.99 | 0.99 | 0.96 | 0.95 | 0.93 | 0.98 |
| Wood | 0.96 | 0.98 | 0.99 | 0.99 | 0.89 | 0.96 | 0.97 | 0.98 | 0.70 | 0.74 | 0.82 | 0.88 |

The presented $F$ values highlight the impact of the transmitter position used for building the training and testing dataset on the prediction performance. The $F$ values for scenario $C_{train}C_{test}$ and scenario $M_{train}M_{test}$ are larger than 0.85 for DT, MP, and RF, and larger than 0.7 for NB, for all the classes, indicating that both the precision and recall are high and all the material classes can be clearly classified when the model is tested on data that correspond to the learned transmitter positions. These results confirm our assumption that the wall material can be classified based on the RE signatures. However, when the transmitter location for the model training differs significantly from those for testing, the classification of the wall materials becomes significantly worse. The ability of the models to generalize well on data that correspond to unknown transmitter positions can be confirmed if the $F$ values for scenario $C_{train}M_{test}$ are satisfactory. The presented $F$ values for scenario $C_{train}M_{test}$ when RF is applied are above 0.85 for glass and wood and above 0.7 for brick and concrete, indicating that glass and wood can be clearly classified with a very small number of $FP$ and $FN$, while brick and concrete can be classified with a slightly larger number of $FP$ and $FN$ compared to glass and wood. For classes glass and wood MP and RF have similar high $F$ values, while for classes brick and concrete the MP has lower $F$ values. Although NB and DT have poorer performance compared to MP and RF, glass and wood still can be distinguished. These results are in line with our previous assumptions based on the reflection coefficient and confirm that the materials with considerably different relative permittivity values can be easily distinguished with all algorithms.

The results show that, even when the model is tested on transmitter positions not included in training set, more than 65% of the instances are correctly classified for NB, more than 70% for DT and MP, and more than 85% for RF. This proves that the model is able to estimate the surface material type (EM properties) using radio links with the transmitter positions which are different than those used for building the model. The obtained classification results confirm the assumptions based on the material's reflection coefficient that materials with different values of relative permittivity can be easily classified. In particular, the differentiation between the materials with considerably different relative permittivities is clearer which suggest that the materials with similar properties

should be grouped together in a class, since they have similar impacts on radio-wave propagation and thus the model cannot (and even does not need to) distinguish them. Finally, the classification results obtained by applying NB, DT, MP, and RF on the same data indicate that the same conclusions regarding the classification of indoor materials can be drawn with different ML algorithms. However, RF brings significant performance improvements, which suggests its use in preference to the other considered algorithms. These findings add substantially to the understanding of surface materials' classification. In addition, they highlight the need to understand the indoor radio channel, the impact of the communication system's setup and the surrounding environment's characteristics on radio propagation for building training datasets that will result in improved predictive models, as well as for an interpretation of the classification results.

## 6. Conclusions and Future Work

In this work we have proposed, formalized, and evaluated a framework for the ML-based wireless sensing of indoor surface material EM properties using RE signatures. In particular, we specified the propagation characteristics to be included in the RE signature, dataset acquisition and storage procedure, ML based radio analyses and performance evaluation, and integration of the domain-knowledge interpretation. The framework has been evaluated on EM material classification in an indoor environment by four ML techniques using different transmitter positions. Results confirm that the proposed methodology can be used for the classification of the surface material in a plain environment and it is promising for more complex indoor environments.

The concept of the pervasive wireless communications will increase the number of radio devices operating in the entire radio spectrum. The distortion of the radio signal does not depend only on its interaction with the environment, but also on the radio transmitter/receiver chains, including antennas, high power amplifiers and housing of the transmitters and receivers. In this respect, the ML is foreseen as an approach to tackle the problem of the wireless sensing of indoor radio environment, in particular due to: (i) huge amount of radio channel state information data in the complete range of radio spectrum supported by location of the transmitter and receiver, and (ii) complexity of the problem, which cannot be solved by traditional mathematical-physical modeling.

We assume that we can prove the proposed concept using datasets obtained by computer simulations and measurements. In the simulated dataset, the signal bandwidth is infinite and the AoA is detected with very high accuracy, even when noise and interference are added to the dataset. In measured datasets, the CIR is usually quantized, resulting in quantization noise. Our preliminary analysis has shown that this has no significant influence on the classification of the wall material. To detect AoA in the measured datasets, which is assumed to be perfect in the simulated datasets, we need to use a multiple antenna system or a massive MIMO approach at the transmitter and receiver, or distribute multiple transmitter–receiver links in the room. The bandwidth of the measurement equipment is a crucial factor in the temporal resolution of the CIR, which affects the estimation of the size and shape of the room. Existing ultra-wideband communication systems have a bandwidth in the range of GHz, resulting in a time resolution of the CIR in the range of tens of centimetres. However, in THz wireless communication, the bandwidth of the communication system is thousands of times larger, resulting in a much better time resolution of the CIR and allowing a more accurate determination of the room properties.

Our investigations in this area still have several open challenges. In future, we will analyze the impact of superimposed errors that exist in real communication system, as well as the indoor space characteristics (geometry, interacting objects, openings, material combinations, surfaces texture) on the proposed approach. In particular, in this work we assumed that radio waves reflect of flat, smooth, and electrically large surfaces and we considered computer generated noiseless RE signature to prove the concept of the proposed framework. Although, in the future, we will additionally consider the scattering effect, and errors introduced by the wireless system, such as limited bandwidth, the non-

linearities introduced by the radio-chains of transmitters and receivers, the Gaussian noise added to the received signal, etc. Analyzing how the superimposed errors affect the proposed concept is difficult, thus we will first evaluate the proposed approach in a controlled environment with generated noise and than we will proceed to the real-life evaluation. For that purpose, we will use radio-propagation data corresponding to many radio links with different position of the end-nodes in versatile indoor spaces, labeled with environment properties, stored in proprietary database. The propagation data includes selection of propagation parameters for many radio-rays between the radio nodes. The indoor environments range from empty, plain rooms, without openings, to complex office and living-space rooms with versatile geometries, with one or several openings for windows and doorways, and equipped with furniture and other objects made of different materials. In that context, we have already built a portable radio-equipment and have started extensive measurement campaigns. At first we look at the office environment measuring the RE in rooms with different shape, size and wall materials.The measured data are than integrated, labeled, and prepared, in the line with a proposed framework, in order to be suitable for ML analysis.

**Author Contributions:** Conceptualization, T.K., T.J., A.Š. and A.H.; methodology, A.H., T.J. and A.Š.; software, T.K.; validation, T.J., A.Š. and A.H.; formal analysis, T.K.; investigation, T.K.; resources, T.K.; data curation, T.K.; writing—original draft preparation, T.K.; writing—review and editing, T.J., A.Š. and A.H.; visualization, T.K., A.H. and T.J.; supervision, A.H.; project administration, A.Š.; funding acquisition, A.Š. All authors have read and agreed to the published version of the manuscript.

**Funding:** This work was supported by the Slovenian Research Agency under grants P2-0016, J2-3048 and J2-2507, and Ad futura programmes for international mobility grand 11011-35/2019-6.

**Acknowledgments:** The authors would like to thank Aleksandra Rashkovska and Dragi Kocev the helpful suggestions regarding the machine learning.

**Conflicts of Interest:** The authors declare no conflict of interest.

## Abbreviations

The following abbreviations are used in this manuscript:

| | |
|---|---|
| 6G | sixth generation |
| ANN | artificial neural network |
| AoA | angle of arrival |
| CTF | channel transform function |
| CIR | channel impulse response |
| CM | confusion matrix |
| CSI | channel state information |
| DT | decision tree |
| EM | electromagnetic |
| $F$ | F-score |
| FCF | frequency coherence function |
| $FP$ | false positives |
| $FN$ | false negatives |
| $HP$ | Horizontal polarization |
| IoT | Internet of Things |
| KNN | k-nearest neighbor KNN |
| LiDAR | light detection and ranging |
| LoS | line of sight |

| ML | machine learning |
| MP | Multilayer Perceptron |
| NB | naive Bayes |
| RE | radio environment |
| RF | random forest |
| RTI | radio tomographic imaging |
| RX | eceiver |
| SLAM | simultaneous localization and mapping |
| SBR | shooting and bouncing ray |
| SVM | support vector machine |
| $TN$ | true negatives |
| $TP$ | true positives |
| TX | transmitter |
| UWB | ultra-wide-band |
| $VP$ | vertical polarization |
| W | wall |
| WSN | wireless sensor network |

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
