# Peer review of "Framework for the Machine Learning Based Wireless Sensing of the Electromagnetic Properties of Indoor Materials"

_electronics, doi:10.3390/electronics10222843_

Round 1
Reviewer 1 Report
This research is well written and organized. The topic is important and the information included is valuable. However there are some minor comments which should be addressed by the authors.
- Keywords: We suggest that the authors should replace keywords such as “electromagnetic properties” and “wireless sensing” because these keywords are already found in the review article title. It is better that they replace them with other keywords to increase the reach of the manuscript.
- Some acronyms should define before use them like RFID, RGB … etc.
- List of References: The references should follow Electronics-MDPI Journal style. For example, some search names in the reference list begin an uppercase letter for each word (such as [1], [2], [3] ... etc.) and others use only an uppercase letter in the first word (such as [4], [6], [8] … etc.), authors should standardize style. The links in references [74] and [81] are not working, please check all the links in list of references. Some references are not very relevant to the topic of the research and are not very necessary. Authors must accurately check all references list to remove all problems.
- English Writing: This article needs moderate proofreading. There are some of grammatical, spelling and typos mistakes, for instance, authors sometimes used “analysed” and other times they used "analyzed", Authors should standardize writing style (UK or USA) … etc. The authors must thoroughly scrutinize the article.
- Paraphrase text: There are many sentences taken from previous articles. Authors must paraphrase all sentences and paragraphs taken from the previous articles. The authors must avoid taking whole sentences from the original articles. They are obligated to paraphrase during the revision of the entire article. Also, authors should include missing references. Plagiarism is not acceptable at all:
- “the offline training phase and the online classification phase. In the offline training phase labeled” (page2) … etc.
- “The tree induction process is non-parametric since it does not require prior assumptions for the probability distribution of the target and other” (page10) … etc.
.
.
.
Etc.
Author Response
Dear Reviewer,
We would firstly like to sincerely thank you for the time and effort you have spent reviewing our original manuscript electronics-1437195, for your thoughtful and constructive feedback, and for the opportunity to revise our manuscript. We believe that your valuable comments helped us to considerably improve the manuscript. The detailed response to your review is in the attached pdf file.
The revised submission includes a clear version of the revised manuscript, and version with marked major, and a version with all tracked changes.
Regards
Tomaž Javornik

Reviewer 2 Report
Please see the attachment.

Author Response

(The authors gave the same response as above.)

Reviewer 3 Report
This paper is dedicated to a conceptual machine learning framework for electromagnetic wave material properties prediction for indoor wireless sensing and communication applications. Problem is actual and many works are available for MRI shielding. Machine learning is bringing the novelty here and must be highlighted in every possible way. Paper is well written and structured.
Some recommendations:
Equation 2, don’t see speed of light there.
Figure 2. Wireless sensing framework architecture. – ML blocks to bolded somehow, to be clearly visible and easily recognizable.
Chapters “4.3. Machine learning based radio-analysis module” and “4.4. Machine learning performance-evaluation module” are very poor. Used ML techniques must be properly introduced with used parameters, NN structures, etc. Probably table with method names and additional data will be useful. Especially for DT, MP, RF. It must be clearly explained what and how is predicted.
Some representation of actual datasets is missing, only training estimations are presented, but how the data looks like? Show just a small piece.
Some additional support on ML ANN/CNN for electromagnetics could be found in: doi: 10.1109/ELECTRONICA50406.2020.9305101.
Author Response

(The authors gave the same response as above.)

Round 2
Reviewer 2 Report
Please see the attachment.

Author Response
Dear reviewer,
Firstly, we would like to sincerely thank you for the additional time and effort you have spent reviewing our revised manuscript. Find attached our response to your comments in the attached file. We try to consider all your suggestions.
Regards
Tomaž Javornik
